# Growth Indicators of Main Species Predict Aboveground Biomass of Population and Community on a Typical Steppe

**DOI:** 10.3390/plants9101314

**Published:** 2020-10-05

**Authors:** Xiaojuan Huang, Yongjie Liu, Niya Wang, Lan Li, An Hu, Zhen Wang, Shenghua Chang, Xianjiang Chen, Fujiang Hou

**Affiliations:** 1State Key Laboratory of Grassland Agro-ecosystems, College of Pastoral Agriculture Science and Technology, Lanzhou University, Lanzhou 730020, China; Huangxj18@lzu.edu.cn (X.H.); yjl@lzu.edu.cn (Y.L.); Wangny2018@lzu.edu.cn (N.W.); lil13@lzu.edu.cn (L.L.); hua09@lzu.edu.cn (A.H.); wangzh17@lzu.edu.cn (Z.W.); cychangsh@lzu.edu.cn (S.C.); chen-x15@ulster.ac.uk (X.C.); 2Key Laboratory of Grassland Livestock Industry Innovation, Ministry of Agriculture, Lanzhou University, Lanzhou 730020, China

**Keywords:** grazing, nondestructive sample, plant height, canopy diameter, interspecific competition

## Abstract

The objective was to explore a fast, accurate, non-destructive, and less disturbance method for predicting the aboveground biomass (AGB) of the typical steppe, by using plant height and canopy diameter of the dominant species, *Stipa bungeana*, *Artemisia capillaris*, and *Lespedeza davurica*, data were observed from 165 quadrats during the peak plant growing season, and the product of plant height (PH) and canopy diameter (PC) were calculated for each species. AGB of population were predicted for the same species and other species through using 2/3 of the measured data, and the optimal predictive equation was linear in terms of determination coefficient. The other 1/3 of the data, which was measured from no grazing paddocks or rotational grazing paddocks, was substituted into the predictive equations for validation. Results showed that PC of one dominant species could be used to predict AGB of the same species or other species well. The predicted and measured values were significantly correlative, and most of the predictive accuracy was above 80%, and not affected by managements of grassland, including rotational grazing or no grazing. A combination of 3 to 6 representative species was used to predict AGB of the community, and the predictive equations with PC of six species as an independent variable were the most optimal because explaining 83.5% variation of AGB. The predictive methods cost 1/15, 1/9, and 1/51 of time, labor, and capital as much as the destructive sample method (quadrat sampling method), respectively, and thus improved the efficiency of field study and protecting the fragile study areas, especially the long-term study sites in grassland.

## 1. Introduction

Grassland is the largest terrestrial ecosystem and one of the three dominant human food production bases in the world [1,2]. As the carrier of elements and energy [3,4], biomass is one of the most important attributes of life systems, and therefore is one of the most necessary measurements especially in the study of life science [5,6]. To address the mechanism for maintaining the structure and function of grassland, the biomass is always measured under various conditions of grazing, the exclusion of livestock, mowing, fertilization, tourism, and so on [7,8]. Although measurement of aboveground biomass (AGB) under different utilization provides a basis for the sustainable management of grasslands [9], a quantitative, accurate, simple, and fast measurement of grassland AGB is still a worldwide problem [10], and there are no enough studies.

The destructive sample method of the vegetation was one of the most predominant for monitoring AGB of global grasslands [11], because simply and easily obtaining accurate data, but required a lot of labor, time, and materials [12,13]. In field studies, frequent sampling caused great interference to the study results and disturbed the study sites, especially in small plots or long-term sites [14,15]. Therefore, empirical prediction modeling was a practical demand because not destroying sites and disturbing the environment, saving time and labor, and simultaneously was easier to be integrated with remote sensing and unmanned aerial vehicle (UAV) [16]. However, AGB of the population was usually predicted by the growth indicators (GI) of the same species in previous studies [17,18,19], seldomly by those of other species, and there were few studies to predict AGB of a community by using GI of dominant species or major accompanying species. At present, for prediction of AGB, most vegetation types were trees and shrubs [20,21,22] and herbaceous plants were rarely studied [23,24]. Moreover, the accuracy of predictive equations depended on the data of destructive samples [25,26,27].

Plant height (PH) and canopy diameter (CD) reflect the vertical and horizontal allocations of AGB, and to a certain extent, the product of PH and CD (PC) represents the plant volume, which illustrates the ability to compete the limited and common resource, i.e., intraspecific competition or interspecific competition [28]. As AGB of population or community is the consequence of both intraspecific and interspecific interactions [29], PH, CD, and PC could reasonably predict AGB (Figure 1). Moreover, several major species always contribute to most of AGB in a community [30], and consequently, their GI could predict the AGB of a community (Figure 1).

Hereby, we conducted a field experiment with no grazing and grazing in a typical steppe of eastern Gansu Loess Plateau. The objectives were to (1) use the PH, CD, and PC of dominant species to predict AGB of the same species or other species, (2) apply the PC of several representative species (dominant species and major accompanying species) to predict AGB of the community, (3) validate the accuracy and stability of the predictive equations under different managements of grassland, including rotational grazing and no grazing. The present study was expected to recommend a fast, nondestructive and accurate method of AGB measurement, which was based on GI of pant and saved the cost of labor, time and funding, and moreover to a certain extent, identified the mechanism of AGB formation and allocation.

## 2. Materials and Methods

### 2.1. Study Site

The study was conducted in Huanxian Grassland Agriculture Trial Station of Lanzhou University, Huanxian County, Gansu Province, China (37.12°N, 106.84°E, 1700 m a.s.l). Mean annual temperature was 8.4 °C and mean annual precipitation was 266.2 mm, over half of which took place from late June through September. The grassland was classified as cool temperate-semiarid temperate typical steppe, abbr. typical steppe [31]. Dominant species of rangeland are *Lespedeza davurica* (Laxm), *Artemisia capillaris* (Thunb), and *Stipa bungeana* (Trin), and three major accompanying species are *Heteropappus altaicus* (Willd) *Potentilla bifurca* (Linn) and *Torularia humilis* (Meyer) (Table 1).

### 2.2. Plot Allocation and Data Collection

In May, 2001, we chose a flat area with similar vegetation type and set up twelve 50 × 100 m paddocks, that consisted of nine rotational grazing paddocks, which had been rotationally grazed by local Tan sheep from early June to early September each year, and three fenced paddocks (no grazing) [32]. After the third cycle of grazing in August from 2001 to 2010, we randomly put four 1 × 1 m quadrats in each paddock and measured the PH and CD of five individual plants and AGB for each species, respectively, and then mean values of each species were calculated for every quadrat. The plant samples were dried in 65 °C oven until constant weight. AGB of community was sum of each species in a quadrat.

### 2.3. Model Establishment and Validation

2/3 of the measured data (76 quadrats) of no grazing paddocks was randomly selected out, and AGB of the same species and other species were predicted with the PC of dominant species as the independent variable. The optimal predictive equation was estimated by the determination coefficient (R^2^) of regressive equations. 1/3 of the measured data in no grazing paddocks (38 quadrats) and rotational grazing paddocks (51 quadrats) were substituted into the predictive equations for validation, respectively. The accuracy and stability were validated according to the total relative error (RS, <10%) (Formula 3), the average absolute value of relative error (RMA, <30%) (Formula 4) and the prediction accuracy (PA, >70%) (Formula 5) [33]. A combination of 3 to 6 species, which were selected out from three dominant species and three major accompanying species (Table 1), were used to predict AGB of community.

AGB of population was predicted by formula 1 as following: (1)$$Y=aX_{i}+b$$
where Y was AGB of the same species and other species, X was PC of the species i, i was the dominant species (i = 1, 2, 3), b is a constant.

AGB of community was predicated by formula 2 as following: (2)$$Y=a_{1}X_{1}+a_{2}X_{2}+\cdots +a_{i}X_{i}+b$$
where Y was AGB of the community, i was the dominant species or accompanying species (i = 1, 2, 3, 4, 5, 6), respectively.

The predictive equation can be validated by formula 3, 4 and 5 as following: (3)$$\mathrm{RS}=[(\sum y_{i}-\sum {\hat{y}}_{i})/\sum {\hat{y}}_{i}]\times 100\%$$
(4)$$\mathrm{RMA}=\;\frac{1}{N}\times \underset{i}{\sum }\frac{\left|y_{i}-{\hat{y}}_{i}\right|}{\left|{\hat{y}}_{i}\;\right|}\times 100\%$$
(5)$$\mathrm{PA}=\left(1-\frac{t_{\alpha }\sqrt{\sum_{i}{\left(y_{i}-{\hat{y}}_{i}\right)}^{2}}}{{\hat{y}}_{i}\sqrt{\mathrm{NN}-T}}\right)\times 100\%$$
where N was number of samples, $y_{i}\;$was the measured value of AGB, ${\hat{y}}_{i}\;$was the estimated value of AGB, $t_{\alpha }\;$was confidence interval.

### 2.4. Statistical Analysis

SPSS 19.0 was used to regress, test, and validate the equations. Through a non-parametric test (K-S test), the data was found to generally follow a normal distribution. The relationship between PH, CD, or PC of three dominant species and AGB of the same species or other species were analyzed by bivariate correlation (Pearson bilateral test) one by one, and if *p* < 0.05, GI of this species was used to establish the predictive equation. Population and community AGB prediction models were established by the the hybrid model (Generalized linearity). The difference of the slope between linear equations was tested by analyzing (comparing mean values) to identify the variation rate of AGB in the horizontal and vertical directions among different species.

## 3. Results

### 3.1. Regression and Validation of Predictive Equations for Population AGB

#### 3.1.1. AGB Predictive Equations and Accuracy Test for the Same Species

PC of three dominant species, *A. capillaris*, *S. bungeana,* and *L. davurica*, could predict AGB of themselves, respectively (Table 2). Among the four kinds of equations, linear one had the largest *R*^2^ and the highest fitting degree, which explained over half variation of AGB. AGB of dominant species rose with an increase in PC. To a certain extent, the slope of linear equation reflects the spatial occupancy of species, and the order of three dominant species was *L. davurica* > *A. capillaris* > *S. bungeana* (*p* = 0.000). 

38 data of no grazing paddocks and 51 of rotational grazing paddocks were substituted into the predictive equations, respectively (Table 3). PC of three dominant species predicted their own AGB at the level of *p* = 0.000, all of RMA and RS were less than 30% and 10%, respectively, and PA was above 83% (Table 3). All of these fell within the allowable error range, which indicated that prediction of PC to AGB of the same species was not affected by different managements of grassland, including no grazing and rotational grazing, and the predictive equations had good accuracy and stability.

#### 3.1.2. Predictive Equations and Accuracy Test for AGB of Other Species

The optimal predictive equations were linear (Table 4). PH of *L. davurica* as independent variable had the highest R^2^ (0.8847) for AGB of *C. hederacea*, and PC of *A. capillaris* had the lowest R^2^ (0.6629) for AGB of *G. verna*. At least 66% variation of AGB can be explained for seven species by GI of dominant species. PH of *L. davurica*, a dominant species of Legumes, could predict AGB of Legumes, Liliaceae and Convolaceae. PH of *S. bungeana*, a dominant species of Gramineae, could predict AGB of Gramineae. AGB of Legumes, Compositae and Gramineae could be predicted by PC of *A. capillaris*, a dominant species of Compositae.

GI data of both no grazing paddocks and grazing paddocks were substituted into predictive equations of other species, and the validations of equations were significant at the level of *p* = 0.000 (Table 5). RMA changed from -32.16% to 29.81%, which were less than 30%, and all RS were less than 10%, which were within the allowable error range of a model. The predictive accuracy of thirteen equations varied between 80% and 90%, while that of five equations varied from 70% to 80%, and that of two equations was over 90%. It identified that predictive equations were not affected by no grazing or rotational grazing, and had enough accuracy and adaptation.

### 3.2. Predictive Equations and Validation for AGB of Community

#### 3.2.1. Establishment of AGB Predictive Equations of Community

As number of species increased from three to six, R^2^ of the predictive equations gradually increased from 0.662 to 0.835 (Table 6). Number of species increased by 1, and the R^2^ of equations averagely increased by 0.0576 (*p* = 0.000). That GI of six species commonly predicted AGB of community could be explained 83.5% variation.

#### 3.2.2. Validation of Predictive Equations for AGB of Community

The estimated values and the measured values were significantly correlated each other at the level of *p* = 0.000 under two managements of grassland (Figure 2). Under no grazing, the estimated values were closer to the measured values than that under rotational grazing. Because the estimated values could be calibrated by regressive equations between themselves and the measured values under both conditions, the predictive equations were not affected by different grassland managements.

## 4. Discussion

### 4.1. PC of Dominant Species Predicting AGB of the Same Species

Despite that GI of all species in alpine meadow were measured to establish the predictive model on the Qinghai-Tibet Plateau [34], the present study only measured GI of dominant species and the accuracy and stability of the prediction equation are not affected or even higher. Compared with that AGB of grasslands was predicted by canopy surface height [35], the present method was not affected by different managements of grassland, and had higher accuracy and stability [36].

### 4.2. GI of Dominant Species Predicting AGB of Other Species

PC of dominant species could predict AGB of other species well, and in this respect, the interspecific interactions between dominant species and other species were positive and the growth of different species were mutual benefit (Table 5), which was different from that the phenomenon of intraspecific facilitation that mainly occurred in greenhouse [37]. Maybe because water was one of the main limiting factors in semiarid regions, plant growth was more sensitive to precipitation and thereby AGB of different species synchronously increased or decreased with change of precipitation. The predictive equations had high accuracy and stability under different managements of grassland because the environmental change was possibly similar to all species and respondence of dominant species were the most sensitive [38]. Therefore, dominant species was reasonable to predict AGB of other species.

### 4.3. GI of Major Species Predicting AGB of Community 

Dominant species contributed most of the biomass to community, and had a greater impact on the ecological process of community than other species [39,40]. AGB of the six representative species (three dominant species and three major accompanying species) accounted for 72% of whole community in present study (Table 1), and therefore, their PC were reasonable to predict AGB of community. Moreover, while using GI of three to five dominant species and major accompanying species to predict community AGB, the accuracy and PA of predictive equations were lower than prediction of six representative species (Table 6). Normally, toxic species were dominant in the degraded grassland, which seldomly were ingested by grazing livestock, and overgrazing altered the environmental factors of grassland far severely than proper grazing [41]. Thereby, predictive equations might need to be modified by the degradation degree of grassland especially under overgrazing, which had took place in about 70% of global grassland in varying degrees [42].

### 4.4. Advantages and Problems of AGB Predictions

Based on a study from 2001 to 2020, the cost of time and labor, and capital input was 1/15, 1/9, and 1/51 of the quadrat sampling method in our study site, respectively (Table 7). The larger the sample size, the more significant the advantage was. Predictive equations were benefit to field study especially in small plots, such as nitrogen addition, warming, precipitation reduction, increased rain or in long-term study. Moreover, remote sensing and UAV technology had been developed to measure plant GI [43,44,45], and thereby could be potentially integrated with the predictive modeling to reduce labor intensity and improve monitoring efficiency. However, due to the difference of precipitation, air temperature, soil, and social environment in different sites [46,47,48], AGB predictive equations of typical steppe should be calibrated before being applied to other types of grassland.

## 5. Conclusions

GI of dominant species was suitable to predict the AGB of the same species and other species. The predictive equation based on GI of three dominant species and three major accompanying species was the optimal for AGB of community in a typical steppe, and the predictive error could also be calibrated by the relationship equations between the observed value and the estimated value. Predictive method of AGB was of great benefit to field study because saving labor and improve efficiency. In the future, accuracy and stability of our predictive equations need to be validated under more managements of grassland and more types of grassland, and following concept of the present study, plant frequency, density, tiller number, growth point density of representative species should be utilized to predict AGB of population and community as well. 

## Figures and Tables

**Figure 1 plants-09-01314-f001:**
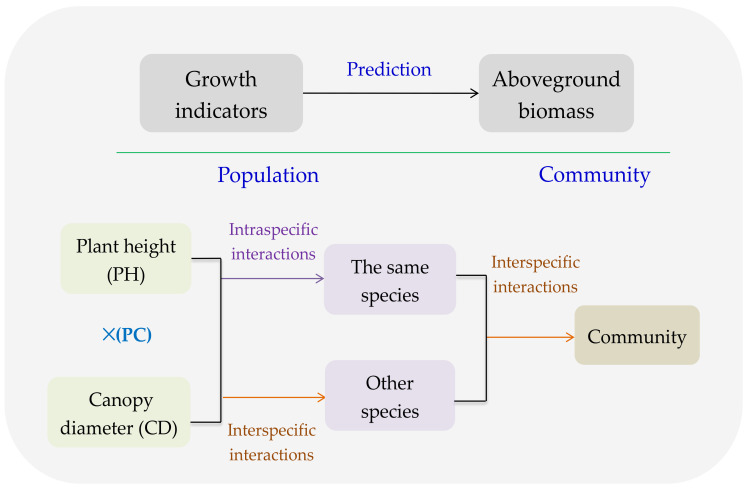
Conceptual sketch for the growth indicators (GI) of plant species predicting the aboveground biomass (AGB) of populations and community.

**Figure 2 plants-09-01314-f002:**
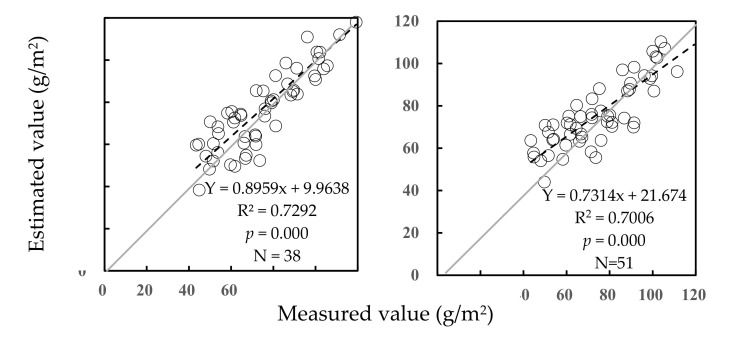
Validation of predictive equations for AGB of community under no grazing (**left**) and rotational grazing (**right**).

**Table 1 plants-09-01314-t001:** Proportion of aboveground biomass (AGB) of species in the community.

Order	Species	Proportion of AGB (%)
1	*Lespedeza davurica*	21.755
2	*Artemisia capillaris*	20.550
3	*Stipa bungeana*	14.587
4	*Heteropappus altaicus*	7.178
5	*Potentilla bifurca*	4.120
6	*Potentilla multifida*	3.476
7	*Torularia humilis*	2.914
8	*Artemisia frigida*	2.891
9	*Oxytropis racemosa*	2.781
10	*Cleistogenes squarrosa*	2.570
11	*Astragalus scaberrimus*	1.930
12	*Ixeridium chinense*	1.818
13	*Astragalus efoliolatus*	0.939
14	*Allium polyrhizum*	0.754
15	*Hedysarum gmelinii*	0.727
16	*Agriophyllum squarrosum*	0.685
17	*Pennisetum centrasiaticum*	0.498
18	*Astragalus galactites*	0.334
19	*Dodartia orientalis*	0.329
20	*Cleistogenes songorica*	0.314
21	*Leymus secalinus*	0.290
22	*Polygala tenuifolia*	0.225
23	*Gueldenstaedtia verna*	0.190
24	*Melilotus officinalis*	0.111
25	*Calystegia hederacea*	0.065
26	*Convolvulus ammannii*	0.033
27	*Cynanchum thesioides*	0.021
28	*Euphorbia esula*	0.020
29	*Convolvulus arvensis*	0.019

**Table 2 plants-09-01314-t002:** Regressive equations of the product of plant height (PH) and canopy diameter (CD) (PC) of dominant species predicting AGB of the same species.

Species	Regressive Equation	R^2^	*p*	F	Sample Size
*L. davurica*	Y = 0.0033x + 0.5354	0.6222	0.000	5.615	76
	Y = 0.1079x^0.4442^	0.3712	0.000	5.615	76
	Y = −4E − 06x^2^ + 0.0043x + 0.4849	0.4233	0.000	5.615	76
	Y = 0.3422ln(x)− 0.6821	0.3106	0.000	5.615	76
*A. capillaris*	Y = 0.0024x + 0.9838	0.5594	0.000	1.529	76
	Y = 0.3352x^0.2768^	0.3621	0.000	1.529	76
	Y = −3E − 06x^2^ + 0.0035x + 0.9048	0.4442	0.000	1.529	76
	Y = 0.335ln(x) − 0.3056	0.3507	0.000	1.529	76
*S. bungeana*	Y = 0.0022x + 0.6248	0.5766	0.000	7.213	76
	Y = −6E − 06x^2^ + 0.0049x + 0.3881	0.4544	0.000	7.213	76
	Y = 0.41ln(x) − 1.0215	0.5126	0.000	7.213	76

**Table 3 plants-09-01314-t003:** Accuracy test of predictive equations under two managements of grassland.

Grassland Management	Dominant Species	Sample Size	Standard Error	R^2^	*p*	F	PA (%)	RMA (%)	RS (%)
No grazing	*L. davurica*	38	0.31	0.814	0.000	8.898	88.26	12.94	2.74
	*A. capillaris*	38	0.15	0.894	0.000	3.957	94.14	6.21	−3.42
	*S. bungeana*	38	0.31	0.907	0.000	4.503	93.47	7.35	2.36
Rotationalgrazing	*L. davurica*	51	0.22	0.780	0.000	2.950	84.25	12.81	4.1
	*A. capillaris*	51	0.12	0.765	0.000	1.831	83.36	10.88	−9.17
	*S. bungeana*	51	0.14	0.807	0.000	3.715	87.54	7.17	−4.35

**Table 4 plants-09-01314-t004:** Regression equations of GI of three dominant species predicting the AGB of other species.

Independent Variable		Predicted Species	Regression Equation	R^2^	*p*	Sample Size
Index	Dominant Species					
PH	*S. bungeana*	*O. racemosa*	Y = 0.1337x − 0.5734	0.6958	0.000	68
	*A. capillaris*	*L. secalinus*	Y = 0.1038x − 0.3787	0.6762	0.000	72
	*L. davurica*	*A. polyrhizum*	Y = 0.0933x − 0.3998	0.6609	0.000	51
	*L. davurica*	*H. gmelinii*	Y = 0.0985x − 0.2896	0.8169	0.000	74
	*L. davurica*	*C. hederacea*	Y = 0.0031x − 0.0146	0.8207	0.000	68
	*L. davurica*	*O. racemosa*	Y = 0.0104x − 0.4065	0.7408	0.000	50
PC	*A. capillaris*	*L. secalinus*	Y = 0.0028x + 0.0075	0.8847	0.000	71
	*A. capillaris*	*I. chinense*	Y = 0.0038x − 0.1941	0.7471	0.000	61
	*A. capillaris*	*H. gmelinii*	Y = 0.0021x + 0.2491	0.7364	0.000	54
	*A. capillaris*	*G. verna*	Y = 0.0017x − 0.0773	0.6629	0.000	59

**Table 5 plants-09-01314-t005:** Accuracy test of predictive equations under two managements of grassland.

	Index	Dominant Species	Other Species	Sample Size	Standard Error	R^2^	*p*	RMA%	RS%	PA%
No grazing	PH	*S. bungeana*	*O. racemosa*	38	0.63	0.886	0.000	12.92	−31.92	88.9
		*A. capillaris*	*L. secalinus*	38	0.34	0.742	0.000	21.82	9.72	79.5
		*L. davurica*	*A. polyrhizum*	38	0.43	0.824	0.000	23.44	−16.55	86.3
		*L. davurica*	*H. gmelinii*	38	0.33	0.799	0.000	18.03	5.57	83.6
		*L. davurica*	*C. hederacea*	38	0.16	0.460	0.000	5.93	−9.51	73.2
		*L. davurica*	*O. racemosa*	38	0.30	0.808	0.000	17.62	−14.92	81.9
	PC	*A. capillaris*	*L. secalinus*	38	0.27	0.913	0.000	15.00	8.58	85.7
		*A. capillaris*	*I. chinense*	38	0.16	0.472	0.000	17.23	4.87	73.1
		*A. capillaris*	*H. gmelinii*	38	0.34	0.807	0.000	16.83	−14.86	84.6
		*A. capillaris*	*G. verna*	38	0.19	0.859	0.000	18.49	−16.22	87.4
Rotational grazing	PH	*S. bungeana*	*O. racemosa*	51	0.24	0.842	0.000	13.98	2.35	84.6
		*A. capillaris*	*L. secalinus*	44	0.27	0.654	0.000	17.80	9.29	77.5
		*L. davurica*	*A. polyrhizum*	49	0.47	0.902	0.000	−32.16	4.73	93.1
		*L. davurica*	*H. gmelinii*	48	0.16	0.589	0.000	−5.64	−6.05	76.8
		*L. davurica*	*C. hederacea*	49	0.06	0.736	0.000	27.45	8.99	87.4
		*L. davurica*	*O. racemosa*	43	0.15	0.918	0.000	−20.57	6.82	93.6
	PC	*A. capillaris*	*L. secalinus*	50	0.12	0.874	0.000	10.26	11.53	89.4
		*A. capillaris*	*I. chinense*	51	0.22	0.637	0.000	13.66	3.08	82.0
		*A. capillaris*	*H. gmelinii*	48	0.18	0.817	0.000	25.36	−7.08	87.7
		*A. capillaris*	*G. verna*	50	0.25	0.684	0.000	29.81	−4.18	81.4

**Table 6 plants-09-01314-t006:** Regressive equations of PC of representative species predicting AGB of community.

Regression Equation	R^2^	F	P
Y = −9.24 + 1.83PC_1_ + 1.291PC_2_ + 0.695PC_3_	0.662	41.621	0.000
Y = 21.293 + 0.093PC_1_ + 0.109PC_2_ + 0.062PC_3_ − 0.03PC_4_	0.697	29.152	0.000
Y = 8.939 + 0.068PC_3_ + 0.151PC_2_+0.086PC_1_ + 0.125PC_4_ − 0.149PC_5_	0.770	30.122	0.000
Y = 17.177 + 0.084PC_1_ + 0.151PC_2_ + 0.055PC_3_ + 0.136PC_4_ − 0.147PC_5_ − 0.135PC_6_	0.835	28.174	0.000

Note: 1 *L. davurica*, 2 *A. capillaris*, 3 *S. bungeana*, 4 *H. altaicu*, *5 P. bifurca*, 6 *P. multifida*.

**Table 7 plants-09-01314-t007:** Comparison between quadrat method and Modeling method

Item	Quadrat Method	Modeling
Duration (day)	15	1
Labor (capita)	9	1
Process	GI measurement, cutting, carrying, drying, weighting, calculation	GI measurement, calculation
Capital input ($)	4776	94

Note: The capital input was calculated on 165 1 × 1 m and the labor cost was daily $22 per capita

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
