# Peer review of "Growth Indicators of Main Species Predict Aboveground Biomass of Population and Community on a Typical Steppe"

_plants, 2020, doi:10.3390/plants9101314_

Round 1

Reviewer 1 Report

In this paper, the authors explored the non-destructive and more accurate methods to predict the above-ground biomass which looks quite impressive. In the general study is a very interesting, well-written draft and easy to follow the contents.

I have only a few minor comments;

L57: What is UAV? I know its corresponds to an unmanned aerial vehicle, but probably reader doesn’t know, However, the authors wrote the full name in L 343-344, which is the end of the manuscript, consider to write the full name in the beginning and later only abbreviate the name.

L 65: Figure 1 can be considered to include a graphical abstract. Figure 1 caption replace “grap” with “graph”.

L 103: The authors didn’t follow the scientific notation through the manuscript. The scientific name should be written in full on the first mention (e.g., Artemisia capillaris) and later always should be genus name abbreviated and full species name (e.g., A. capillaris).

Follow the scientific notation throughout the manuscript since the authors used the full name whenever mentioned in the manuscript.  For example, in L 103, it should be the genus name abbreviated and the species name should be written in full since it been already mentioned in full in L 78-79.

L 120: Replace the “abovementioned” with “above-mentioned” throughout the manuscript.

L 128: Throughout the manuscript, authors refer to the statistically significant level “..at the level of 0.05…” I would suggest considering mentioning P values alongside for example, “..at the level of P = 0.05”. Consider changing this throughout the manuscript.

L 142: Table 1 footnotes, the scientific names should be in italics.

L 148-149: What are RMA and RS? The authors mentioned full name in the figures captions but not in the text, since its the first time mentioned in the text, consider writing in full.

L 167: Explain the RMA and RS name in the table footnote.

L 206: Replace “table 6” with “Table 6”.

L 214: replace “Gueldenstaedtia Verna” with “Gueldenstaedtia verna”.

L 233: replace “P < 0.001” with “P < 0.05”.

L 260: Replace “figure. 3” with “Figure 3”.

L 323: Table 8 is not cited in the text throughout the manuscript.

Author Response

28 September 2020

Editors

Plants

REF: Growth Indicators of Dominant Species Predict Aboveground Biomass of Population and Community on Typical Steppe.

Dear Editors,

Thank you very much for your decision with three reviewer’s comments. We really appreciate your constructive and thoughtful comments and suggestions, following which we have significantly improved our manuscript. We have identified these in the Acknowledgement section. we have considered all their comments very carefully and revised the manuscript accordingly. We highlighted our revision in Yellow in the manuscript for your and reviewers’ convenience.

We hereby the follow reviewers’ comments in order and explain what have done point by point as follows.

Comments and Suggestions for Authors from Reviewer 2

Point 1: Line 17. CH means?

Response 1: Thank you for your comments. We re-edited the abbreviation of the full text, changing the original CH to the current PC, which represents the product of plant height and canopy diameter.

Point 2: Line 44. “This…” refers to biomass, it is not clear.

Response 2: Thanks for your comments. We have explained this in lines 41-43.

Point 3: Line 57.UAV means? Really its meaning appears in the last line of the manuscript.

Response 3: Thanks for your specific comment. We have explained the UAV in line 52

Point 4: Figure 1.Which new information adds this Figure to the text explanation? Perhaps may be deleted or modified.

Response 4: Thank you for your comments. We have explained Figure 1 in lines 73-76.

Point 5: Line 79. A more information about communities, botanical and diversity index should be added, if not, the meaning of the research results is incomplete.

Response 5: Many thanks for your directions! We have added Table 1 to the text in line 85, and the table contains indicators such as diversity index and communities suggested by reviewers.

Point 6: 2.2. Section (Plot allocation). A bit of confusion run through the sampling methodology description. Perhaps a Figure and a Tables with plots allocation and the variables estimated can help to do a better understanding of the field sampling procedure. In this section are two paragraphs, one refers to community and the second to population, but this identification appears at the end of each one. By other hand, lot of same information are repeated.

Response 6: Thank you for your comments. We've changed it in lines 87-94 and removed the duplicates.

Point 7: Line 110. Why the selection of this number of plots: 76 and 38?     

Response 7: Thanks for your comments. We have explained this in lines 98-103

Point 8: Section 2.4.oined the Statistical procedure explanation to the Section before. Really the only new information is the use o Excel software package, that, really is not a Statistical software…

Response 8: Thank you very much for your advice. We have refined and modified the statistical methods in lines 118-124 .

Point 9: Line 143.”the” must be The.

Response 9: Thanks. We have changed “the” to “The” .

Point 10: Table 2. Sort the Dominant population items (1B, 2B, 3B) in the same way as in the Table 1.

Response 10: We have added Table 1 and Table 2 and merged some tables. The previous Table 2 is changed into Table 3, and the corresponding modifications have been made in line 135-136 according to expert opinions.

Point 11: Line 159. “Super species” means?

Response 11: We thank reviewer very much for your careful directions. We have changed “Super species” to “dominant species”.

Point 12: Line 174. Put only the items that appears in the Table, H and CH (not HC).  C, as independent variable, does not appears. The same for the other Tables. Each Table should be explained, at all, by itself. e.g.Table 5 does not show the meaning of 3,2,1….

Response 12: Many thanks for your directions, We have modified the full-text table based on expert advice and deleted the indicators irrelevant to the chart.

Point 13: Lines 206-2014. Please, do not repat the same information that are show in the Table. It may be more useful do a more worked data presentation, e.g. presents the results classified into a “prediction classes” 90 %/ 85-90/ 75-85…

Response 13: Thank you for your comments. We have rewritten this paragraph in line 155-163 accordance.

Point 14: Line 221.3, 4, 5, 6 dominant species, from how many community species. This information is essential to understand the power of the prediction model, and the community description should be recorded in M and M. section.

Response 14: thank you for your advice. We have added Table 1 in line 85.

Point 15: Line 222. Why top six species, and not four top our seven top species.

Response 15: Many thanks for your directions. We have explained this problem in lines 221-227.

Point 16: Table 7 Please, follow the same pattern in table 1 and 7: N, P, F.

Response 16: We thank reviewer very much for your careful directions. We have made changes to Table 7.

Point 17: Lines 229-237 In order to be able to determine the really significance of the Prediction model, it would very interesting show the biomass ratio between 3,4,5,6 dominant species versus the whole plant community. for that, and as I said before, a complete plant community description should be done.

Response 17: Thank you for your comments. We have added Table 1 in line 85.

Point 18: Line 240. “the ordinate was then taken as the ordinate”. Of course, it should be a lapsus.

Response 18: Thank you for your kindly remind. We have fixed this error.

Point 19: 4.4 Section In the same way that you show in this section, you could stablish the difference of cost for, e. g., use 3 or 6 dominant species in the prediction model?

Response 19: Thank you for your constructive comments. We have explained in 221-227.

Reviewer 2 Report

Dear authors, your work is interesting and provide predictions on aboveground biomass. There are some changes that need to be addressed in order to improve your work and make it more discoverable for the researchers in this field.

L15-32 Abstract.

Please add a sentence as a background for your research.

CH - when first used in text provide the full name not only the abbreviation.

You better remove the regression equation from the abstract and discuss it.

Make all the sentences short. Avoid long sentences.

L47 - Add more references

L68-69 - Rewrite the third objective.  To verify the accuracy and stability of the created model ....

L71 - Change the mane of figure - Experimental design or Research Framework

L78-79Stipa bungeana, Artemisia capillaris and Lespedeza bicolor. Add Author for each species

Discussion section - Add more references. You need to sustain better your findings and compare with more works.

Conclusion. Rewrite the conclusions in shorter sentences, in order to point the most important findings and result. Remove the equation, and better present each coefficient and it`s importance. In this way you will promote your model.

Author Response

(The authors gave the same response as above.)

Reviewer 3 Report

 It is a very interesting and useful research (scientific soundness), but the manuscript needs some important changes, mainly in Material and Methods section in order to understand in a properly way the true significance of the results obtained.

REMARKS

Line 17.                  CH means?

Line 44.                “This…” refers to biomass, it is not clear.

Line 57.                UAV means? Really its meaning appears in the last line of the manuscript.

Figure 1.               Which new information adds this Figure to the text explanation? Perhaps may be deleted or modified.

Line 79.                A more information about communities, botanical and diversity index should be added, if not, the meaning of the research results is incomplete.

2.2. Section (Plot allocation).          A bit of confusion run through the sampling methodology description. Perhaps a Figure and a Tables with plots allocation and the variables estimated can help to do a better understanding of the field sampling procedure. In this section are two paragraphs, one refers to community and the second to population, but this identification appears at the end of each one. By other hand, lot of same information are repeated.

Line 110.                                Why the selection of this number of plots: 76 and 38?      

Section 2.4.           Joined the Statistical procedure explanation to the Section before. Really the only new information is the use o Excel software package, that, really is not a Statistical software…

Line 143.               ”the” must be The.

Table 2.  Sort the Dominant population items (1B, 2B, 3B) in the same way as in the Table 1.

Line 159.               “Super species” means?

Line 174.               Put only the items that appears in the Table, H and CH (not HC).  C, as independent variable, does not appears. The same for the other Tables. Each Table should be explained, at all, by itself.  e.g.   Table 5 does not show the meaning of 3,2,1….

Lines 206-2014.                  Please, do not repat the same information that are show in the Table. It may be more useful do a more worked data presentation, e.g. presents the results classified into a “prediction classes” 90 %/ 85-90/ 75-85…

Line 221.       3, 4,5,6 dominant species, from how many community species. This information is essential to understand the power of the prediction model, and the community description should be recorded in M and M. section.

Line 222.               Why top six species, and not four top our seven top species.

Table 7                 Please, follow the same pattern in table 1 and 7: N, P, F

Lines 229-237     In order to be able to determine the really significance of the Prediction model, it would very interesting show the biomass ratio  between 3,4,5,6 dominant species  versus  the whole plant community . for that, and as I said before, a complete plant community description should be done.

Line 240.              “the ordinate was then taken as the ordinate”. Of course, it should be a lapsus.

4.4 Section          In the same way that you show in this section, you could stablish the difference of cost for, e.g., use 3 or 6 dominant species in the prediction model?

Author Response

28 September 2020

Editors

Plants

REF: Growth Indicators of Dominant Species Predict Aboveground Biomass of Population and Community on Typical Steppe.

Dear Editors,

Thank you very much for your decision with three reviewer’s comments. We really appreciate your constructive and thoughtful comments and suggestions, following which we have significantly improved our manuscript. We have identified these in the Acknowledgement section. we have considered all their comments very carefully and revised the manuscript accordingly. We highlighted our revision in Yellow in the manuscript for your and reviewers’ convenience.

We hereby the follow reviewers’ comments in order and explain what have done point by point as follows.

Comments and Suggestions for Authors from Reviewer 3

Point 1: L15-32 Abstract. Please add a sentence as a background for your research.

Response 1: Thank you for your comments. We have refined the summary and added background notes in lines 15-30.

Point 2: CH - when first used in text provide the full name not only the abbreviation.

Response 2: Thank you for your comments. We have reviewed the full text and changed the abbreviations that first appeared in the article to full names.

Point 3: You better remove the regression equation from the abstract and discuss it.

Response 3: Thank you for your comments. We have removed the regression equation from the abstract and discussed it in lines 217-227.

Point 4: Make all the sentences short. Avoid long sentences.

Response 4: Thanks. We have adjusted the sentence structure of the whole text.

Point 5: L47 - Add more references.

Response 5: Thank you, we have added reference in the manuscript.

Point 6: L68-69 - Rewrite the third objective. To verify the accuracy and stability of the created model....

Response 6: Thank you for your comments. We have modified and improved the third objective in lines 69-73.

Point 7: L71- Change the mane of figure - Experimental design or Research Framework.

Response 7: Thank you for your advice .We have changed the title of Figure 1 in lines 75-76.

Point 8: L78-79Stipa bungeana, Artemisia capillaris and Lespedeza bicolor. Add Author for each species

Response 8: Thanks for your specific comments. We have added the authou fou three dominant species in line 104-105.

Point 9: Discussion section - Add more references. You need to sustain better your findings and compare with more works.

Response 9: Thanks. We have revised the discussion section of the article according to the experts' Suggestions and added more references.

Point 10: Conclusion. Rewrite the conclusions in shorter sentences, in order to point the most important findings and result. Remove the equation, and better present each coefficient and it`s importance. In this way you will promote your model.

Response 10: Thank you for your comments. We have revised the conclusions according to expert opinions in line243-247.

Round 2

Reviewer 3 Report

Changes were done in a properly way

This manuscript is a resubmission of an earlier submission. The following is a list of the peer review reports and author responses from that submission.

Round 1

Reviewer 1 Report

This an interesting study about the use of growth index for biomass production of steppe plant communities. Your study is well designed and the use of growth index to calculate biomass production for grassland plant populations and communities might serve for several applications. In its present form, however, I miss a strong implication for conservation or/and sustainabilty (e.g. optimized land use). Without this implication the manuscript does not fit for publication in Sustainabilty, I therefore recommend to submit it to a Journal with focus vegetation ecology or vegetation science.

Reviewer 2 Report

Dear authors, the subject is interesting and the article provide a good scientific basis for the study of grasslands. I was pleased to read it. There are some points that need to be addressed in order to improve your work. 

L2 - change predicating with predicting in title

L14-L29 - Abstract

Pay attention to the entire construction of abstract. There are some missing points as end of sentence (L28) and the length of sentences produce some confusion. You need to rewrite all the sentences in order to have shorter ones and better point your ideas and findings.

L34-37 - Split in 2 sentences

L38 - the biomass of grassland - you repeat the word in same sentence

L41-43 - rewrite the sentence, is confusing

L44 - biomass of grasslands

L57 - change: at home and abroad. It does not sound scientific

L62 - breadth - do you mean amplitude, width or spread?

L63 - above-moentioned - do you mean aboveground?

L57-67 - This part of the article provide informations about your experimental basis and concept. You need to better write your each hypothesis or objectives. They are not clear. Please check the english. 

L64 - Our results will not only reduce labor.

L302 - change baked to something like dried

L89 - table 1 - change 0.000 with p<0.001

L90 all latin names with italic, make this change in all text

For all tables. It is better to put all the latin names (eventually reduce to L.b. - Lespedeza bicolor as an example, and A.c., S.b.) instead of numbers 1, 2, 3, 4 etc. It will be easier to read and understand the tables. 

Entire result section - when you have your own regression and equations please make sure that they not occupy a large area in text and do not present them one after another one. Compare them in different sentences but not in the same line.

Add some more references in the discussion section.

Reviewer 3 Report

Main question

In this actual form, the text is quite difficult to following, e.g. Material and Methods appears in section 4., before Conclusions and after Discussion?

The text is so confused too, and it is not clear when the authors talk about biomass prediction for population or community.

There is not a minimal botanical description of the community studied, neither the sampling method nor plant measure methodology.

Not all the prediction models are suitable, e.g., Table 4, 2H-9B / 2HC-4B presents a lower R2, or high relative error 3H-7B/ 2H-9B (Table 5), similar in Table 6 (2HC-9B). These different results should be explained and include into the Discussion section, in order to justify the selection or “erasing” of the each prediction  model.

Other questions

Table 1, Table 4, Table 7 shows the R2 values, but the other Tables show R values?

Species names are “wrong writing” in some parts of the manuscript, e.g., Table 1/ Table 4 footnote (italic font must be used, and specific name too).

Conclusion

The aim of the research is very interesting and scientific soundness,  but a more  work about methodology description, a more in deep discussion about the results founded, and its applicability should be done, e.g, Which is the  relationship between the research results showed and  remote sensing and unmanned  aerial vehicle technology (line 281).

And of course, not all the prediction models are really suitable, and a more discussion and correspondent conclusions should be development in order to explain and identified the “better models” and the conditions for their applicability.

Formal questions (sections ordered in the text, species scientific names writing, Table information, clear differentiation between population and community predictions…) should be attendant in a serious way.